# Combination and competition between path integration and landmark navigation in the estimation of heading direction

**Sevan K. Harootonian**[1], **Arne D. Ekstrom**[2,3], **Robert C. Wilson**[2,3,4]*

**1** Department of Psychology, Princeton University, Princeton, New Jersey, United States of America,
**2** Department of Psychology, University of Arizona, Tucson, Arizona, United States of America, **3** McKnight Brain Institute, University of Arizona, Tucson, Arizona, United States of America, **4** Cognitive Science Program, University of Arizona, Tucson, Arizona, United States of America

* bob@arizona.edu

**Data Availability Statement:** All data and code to create the models are available publicly at: https://github.com/sharootonian/

## Abstract

Successful navigation requires the ability to compute one's location and heading from incoming multisensory information. Previous work has shown that this multisensory input comes in two forms: body-based idiothetic cues, from one's own rotations and translations, and visual allothetic cues, from the environment (usually visual landmarks). However, exactly how these two streams of information are integrated is unclear, with some models suggesting the body-based idiothetic and visual allothetic cues are combined, while others suggest they compete. In this paper we investigated the integration of body-based idiothetic and visual allothetic cues in the computation of heading using virtual reality. In our experiment, participants performed a series of body turns of up to 360 degrees in the dark with only a brief flash (300ms) of visual feedback *en route*. Because the environment was virtual, we had full control over the visual feedback and were able to vary the offset between this feedback and the true heading angle. By measuring the effect of the feedback offset on the angle participants turned, we were able to determine the extent to which they incorporated visual feedback as a function of the offset error. By further modeling this behavior we were able to quantify the computations people used. While there were considerable individual differences in *performance* on our task, with some participants mostly ignoring the visual feedback and others relying on it almost entirely, our modeling results suggest that almost all participants used the same *strategy* in which idiothetic and allothetic cues are combined when the mismatch between them is small, but compete when the mismatch is large. These findings suggest that participants update their estimate of heading using a hybrid strategy that mixes the combination and competition of cues.

## Author summary

Successful navigation requires us to combine visual information about our environment with body-based cues about our own rotations and translations. In this work we investigated how these disparate sources of information work together to compute an estimate

CombinationAndCompetitionHeadingDirection DOI: 10.5281/zenodo.5879723.

**Funding:** Research supported by grants from NSF Division of Behavioral and Cognitive Sciences [BCS-1630296] awarded to Arne Ekstrom. The funders had no role in study design, data collection and analysis, decision to publish, or preparation of the manuscript.

**Competing interests:** The authors have declared that no competing interests exist.

of heading. Using a novel virtual reality task we measured how humans integrate visual and body-based cues when there is mismatch between them—that is, when the estimate of heading from visual information is different from body-based cues. By building computational models of different strategies, we reveal that humans use a hybrid strategy for integrating visual and body-based cues—combining them when the mismatch between them is small and picking one or the other when the mismatch is large.

## Introduction

The ability to navigate—to food, to water, to breeding grounds, or even to work—is essential for survival in many species. To navigate effectively we need to continuously update our estimates of location and heading in the environment from incoming multisensory information [1–3]. This multisensory input comes in two forms: idiothetic cues, from one's own rotations and translations (including body-based cues from the vestibular, proprioceptive, and motor efferent copy systems, as well as visual optic flow), and allothetic cues, from the environment (usually visual landmarks). In this paper we investigate how information from body-based idiothetic and visual allothetic cues are integrated for navigation.

Navigation using <u>only</u> idiothetic cues (for example navigating in the dark) is called Path Integration. Path Integration is notoriously inaccurate involving both systematic and random errors [4–6]. For example, systematic error include biases induced by execution and past experiences such as history effects from past trials [7–10]. Random errors include noise in the body-based idiothetic sensory cues as well as in the integration process itself. These random errors accumulate with the square root of the distance and duration traveled in a manner similar to range effects in magnitude estimations; a consequence of the Weber–Fechner and Stevens' Power Law [10–14]. Despite these sources of errors in path integration, humans and animals rely heavily on path integration because body-based idiothetic cues are constantly present (unlike visual allothetic landmark cues that may be sparse [6, 15]). In addition, path integration allows for flexible wayfinding by computing a route through new never experienced paths, and adjust for unexpected changes along the way [4, 16, 17].

Navigation using visual allothetic cues (for example navigating a virtual world on a desktop computer) using landmarks is called Map or Landmark Navigation [1, 18]. Pure landmark navigation (i.e. without body-based idiothetic cues) can only be studied in virtual environments, where body-based idiothetic cues can be decoupled from visual allothetic cues. In these studies, human participants show no differences in their navigational ability with or without isolation from body-based idiothetic cues, emphasizing that landmark navigation is a separate, and potentially independent computation from path integration [19].

Navigation using <u>both</u> body-based idiothetic and visual allothetic cues relies on both path integration and landmark navigation, yet exactly how the two processes work together is a matter of debate. In 'cue combination' (or 'cue integration') models, independent estimates from path integration and landmark navigation are combined to create an <u>average</u> estimate of location and heading. This averaging process is often assumed to be Bayesian, with each estimate weighed according to its reliability [20, 21]. Conversely, in 'cue competition' models, estimates from path integration and landmark navigation compete, with one estimate (often the more reliable) overriding the other completely. Based on this view, Cheng and colleagues proposed that path integration serves as a back-up navigational system that is used only when allothetic information is unreliable [22].

Empirical support exists for both cue combination and cue competition accounts. In a study by Chen and colleagues [23], humans in a virtual-navigation task averaged estimates from path integration and landmark navigation according to their reliability, consistent with a Bayesian cue combination strategy. Conversely, in a similar experiment by Zhao and Warren [24], participants primarily used visual allothetic information, often ignoring body-based idiothetic cues even when the mismatch was as large as 90˚, consistent with a cue competition strategy. Similar discrepancies exist across the literature, with some studies supporting cue combination (and even optimal Bayesian cue combination) [23, 25–28], and others more consistent with cue competition [24, 29–34].

Further complicating these mixed findings across studies are the large individual differences in navigation ability between participants [2, 35–37]. These individual differences encompass both high level processes, such as learning, knowledge, and decisions about routes [38–40], as well as lower level processes, such as how individuals respond to Corriolis forces and the perception of angular rotations due to differences in semi-circular canal radii [41, 42]. Such large individual differences also impact the integration of body-based idiothetic and visual allothetic cues and may be one reason for the discrepancies in the literature [23, 24].

In this paper, we investigate how people combine body-based idiothetic and visual allothetic cues in the special case of computing egocentric head direction. We focus on head direction because of its relative simplicity (compared to estimating both heading and location) and because the head direction system is known to integrate both vestibular (idiothetic) and visual (allothetic) cues [43]. In our task, participants performed full-body rotations to a goal with only a brief flash of visual feedback that either matched or mismatched their expectations. By building models of this task that capture the key features of cue combination and cue competition strategies, as well as the 'pure' strategies of path integration and landmark navigation, we find evidence for a hybrid strategy in which the estimates of path integration and landmark navigation are combined when the mismatch is small, but compete when the mismatch is large. Model comparison suggests that almost all participants use this strategy, with the large individual differences between participants being explained by quantitative differences in model parameters not qualitative differences in strategy. We therefore suggest that this flexible, hybrid strategy may underlie some of the mixed findings in the literature.

## Methods

### Ethics statement

All participants gave written informed consent to participate in the study, which was approved by the Institutional Review Board at the University of Arizona.

### Participants

33 undergraduate students (18 female, 15 male, ages 18–21) received course credit for participating in the experiment. Of the 33, 3 students (3 female) did not finish block 1 due to cybersickness and were excluded from this study.

### Stimuli

The task was created in Unity 2018.4.11f1 using the Landmarks 2.0 framework [44]. Participants wore an HTC Vive Pro with a wireless Adapter and held pair of HTC Vive controllers (Fig 1A). The wireless headset, that was powered by a battery lasting about 2 hours, was tracked using 4 HTC Base Station 2.0, which track with an average positioning error of 17mm with 9 mm standard deviation [45]. Participants were placed in the center of a large

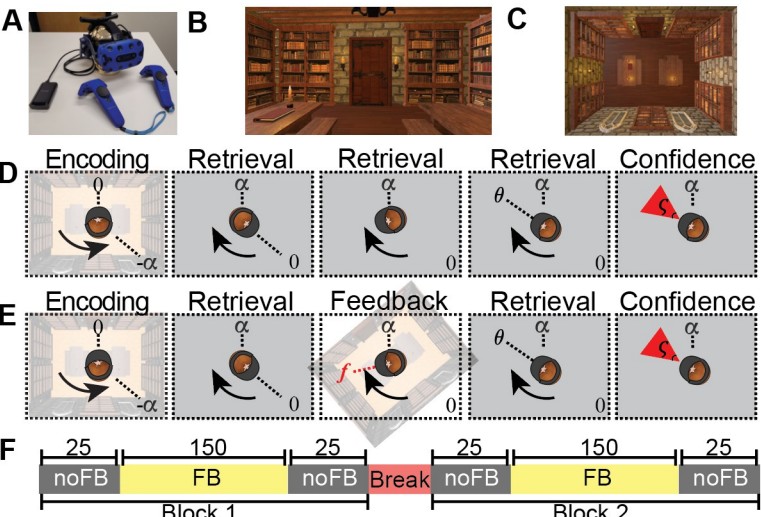

**Fig 1. Task procedure.** (A) Participants wear an HTC VIVE headset along with the handheld controllers to immerse themselves in a virtual room (B, C). First person view of the virtual environment at the beginning of a trial. (C) Top down view of the virtual environment. (D, E) Trial timeline for No Feedback (D) and Feedback (E) trials. At the start of each trial they face the door of the room and turn through $-\alpha$ degrees with visual feedback present. Visual feedback is then removed (gray squares) and they must turn back $\alpha$ degrees to face the door again. At the end of the turn participants stop at heading angle $\theta_t$ and report their confidence $\varsigma$ by adjusting the size of a red rectangle. The only difference between the No Feedback and Feedback conditions is the presence of a brief flash of visual information part way through the turn in the Feedback condition (E). Overall participants completed 300 trials of the Feedback condition and 100 trials of the No Feedback condition over the course of the experiment (F).

rectangular (13m x 10m x 5m) naturalistic virtual room with several decorations (Fig 1B). This was to ensure that visual feedback from different angles would be distinguishable by the geometry of the room and the decorations. We controlled the saliency of the decorations for left and right-handed rotations by placing identical objects in similar vertical positions (Fig 1C).

## Preprocessing and exclusion criteria

The 30 participants included for data analysis all completed block 1 (Fig 1F). Due to tracking failures in the headset caused by a low battery, data from 3 participants was lost for most of session 2. Nevertheless we include data from all 30 participants in our analysis. Trials were removed if participants rotated in the wrong direction or if they pressed in the incorrect button to register their response. We also removed trials in which participants responded before they received feedback.

## The rotation task

During the procedure, participants were first guided, with a haptic signal from hand held controllers, through a rotation of $-\alpha$ degrees with visual feedback present (encoding phase). They were then asked to turn back to their initial heading, i.e. to turn a 'target angle' $+\alpha$, with visual feedback either absent or limited (retrieval phase). In the No Feedback condition, participants received no visual feedback during the retrieval phase. In the Feedback condition, participants received only a brief flash of (possibly misleading) visual feedback at time $t_f$. By quantifying the extent to which the feedback changed the participants' response, the Rotation Task allowed us to measure how path integration is combining with visual feedback to compute heading.

More precisely, at the start of each trial, participants faced the door in a virtual reality room (Fig 1B and 1C) and were cued to turn in the direction of the haptic signal provided by a

controller held in each hand (Fig 1A). Haptic signals from the controller held in the left hand cued leftward rotations (counterclockwise), while signal from the right controlled cued participants to rotate rightward (clockwise). Participants rotated until the vibration stopped at the encoding angle, $-\alpha$, which was unique for each trial (sampled from a uniform distribution, $\mathcal{U}(\alpha)$). Participants were free to turn at their own pace and the experimenter provided no guidance or feedback on their rotational velocity. During the encoding procedure, participants saw the virtual room and were able to integrate both visual and vestibular information to compute their heading.

During the retrieval phase, participants had to try to return to their original heading direction (i.e. facing the door) with no (No Feedback condition) or limited (Feedback condition) visual feedback. At the beginning of the retrieval phase, participants viewed a blank screen (grey background in Fig 1D). They then attempted to turn the return angle, $+\alpha$, as best as they could based on their memory of the rotation formed during encoding. Participants received no haptic feedback during this retrieval process.

The key manipulation in this study was whether visual feedback was presented during the retrieval turn or not and, when it was presented, the extent to which the visual feedback was informative. In the No Feedback condition, there was no visual feedback and participants only viewed the blank screen—that is they could only rely on path integration to execute the correct turn. In the Feedback condition, participants saw a quick (300ms) visual glimpse of the room, at time $t_f$ and angle $f$, which was either consistent or inconsistent their current bearing $\theta_{t_f}$. Consistent feedback occurred with probability $\rho = 70\%$. In this case the feedback angle was sampled from a Gaussian centered at the true heading angle and with a standard deviation of 30˚. Inconsistent feedback occurred with probability $1 - \rho = 30\%$. In this case the feedback was sampled from a uniform distribution between $-180$˚ and $+180$˚. Written mathematically, the feedback angle, $f$ was sampled according to

$$f \sim \begin{cases} \mathcal{N}(f|\theta_{t_f}, \sigma_f^2) & \text{with probability } \rho = 0.7 \\ \\ \mathcal{U}(f) & \text{with probability } 1 - \rho = 0.3 \end{cases} \tag{1}$$

where $\mathcal{N}(f|\theta_{t_f}, \sigma_f^2)$ is a Gaussian distribution over $f$ with mean $\theta_{t_f}$ and standard deviation $\sigma_f = 30$˚.

This form for the feedback sets up a situation in which the feedback is informative enough that participants should pay attention to it, but varied enough to probe the impact of misleading visual information across the entire angle space. To further encourage participants to use the feedback, they were not told that the feedback could be misleading.

Upon completing the retrieval turn, participants indicated their response with a button press on the handheld controllers (Fig 1D), thus logging their response angle, $\theta_t$. Next, a red triangle appeared with the tip centered above their head and the base 6 meters away. Participants then adjusted the angle $\varsigma$ to indicate their confidence in their response angle using the touch pad on the controllers. In particular, they were told to adjust $\varsigma$ such that they were confident that the true angle $\alpha$ would fall within the red triangle (Fig 1D). Participants were told they would received virtual points during this portion, with points scaled inversely by the size of the $\varsigma$ such that a small $\varsigma$ would yield to higher points (risky) and large $\varsigma$ would yield to lower points (safe).

After completing their confidence rating, the trial ended and a new trial began immediately. To ensure that participants did not receive feedback about the accuracy of their last response, each trial always began with them facing the door. This lack of feedback at the end of the trial

ensured that participants were unable to learn from one trial to the next how accurate their rotations had been.

Overall the experiment lasted about 90 minutes. This included 10 practice trials (6 with feedback, 4 without) and 400 experimental trials split across two blocks with a 2–10 minute break between them (Fig 1E). Of the 200 trials in each block, the first and last 25 trials in each block were No Feedback trials, while the remaining 150 were feedback trials. Thus each participant completed 100 trials in the No Feedback condition and 300 trials in the Feedback condition.

Participants were allowed to take a break at any time during the task by sitting on a chair provided by the experimenter. During these breaks participant continued to wear the VR headset and the virtual environment stayed in the same egocentric orientation.

## Models

We built four models of the Rotation Task that, based on their parameters, can capture several different strategies for integrating visual allothetic and body-based idiothetic estimates of location. Here we give an overview of the properties of these models, full mathematical details are given in Section 1 of S1 File. To help the reader keep track of the many variables, a glossary is given in Table 1. Note, the following models focus exclusively on the Retrieval portion of the task. The target location from the Encoding portion of the task is modeled in Section 2 of S1 File.

**Path Integration model.** In the Path Integration model we assume that the visual feedback is either absent (as in the No Feedback condition) or ignored (as potentially in some participants). In this case, the estimate of heading is based entirely on path integration of body-based idiothetic cues. To make a response, i.e. to decide when to stop turning, we assume that participants compare their heading angle estimate, computed by path integration, with their memory of the target angle. Thus, the Path Integration model can be thought of as comprising two processes: a path integration process and a target comparison process (Fig 2).

In the path integration process, we assume that participants integrate scaled and noisy idiothetic cues about their angular velocity, $d_t$. These noisy velocity cues relate to their true angular velocity, $\delta_t$ by

$$d_t = \gamma_d \delta_t + v_t \tag{2}$$

where $\gamma_d$ denotes the gain on the velocity signal, which contributes to systematic under- or over-estimation of angular velocity and $v_t$ is zero-mean Gaussian noise with variance that increases in proportion to the magnitude of the angular velocity, $|\delta_t|$, representing a kind of Weber–Fechner law behavior [10].

We further assume that participants integrate this biased and noisy velocity information over time to compute a probability distribution over their heading. For simplicity we assume this distribution is Gaussian such that

$$p(\theta_t | d_{1:t-1}) = \mathcal{N}(\theta_t | m_t, s_t^2) \tag{3}$$

where the $m_t$ is the mean of the Gaussian over heading direction and $s_t^2$ is the variance. Full expressions for $m_t$ and $s_t^2$ are given in the Section 1 of S1 File. Fig 2 illustrates how this distribution evolves over time.

In the target comparison process, we assume that participants compare their estimate of heading from the path integration process to their memory of the target angle. As with the encoding of velocity, we assume that this memory encoding is a noisy and biased process such

**Table 1. Glossary of key variables in the paper.**

| | |
|---|---|
| **Observed or controlled by experimenter** | |
| $t$ | time step during rotation |
| $\theta_t$ | true heading angle at time $t$ |
| $\alpha$ | true target angle |
| $\delta$ | true velocity |
| $f$ | feedback angle |
| $t_f$ | time of feedback |
| $\sigma_f$ | standard deviation of true feedback |
| $\rho$ | generative probability that the feedback is true on any trial (= 70%) |
| $\varsigma$ | confidence reported by subject |
| **Observed, used, or computed by participant** | |
| *Estimates of heading* | |
| $m_t$ | estimated heading from Path Integration |
| $s_t$ | uncertainty in Path Integration estimate of heading |
| $s_0$ | subject's initial uncertainty in heading angle |
| $\hat{m}_t$ | estimated heading from Kalman Filter |
| $\hat{s}_t$ | uncertainty in Kalman Filter estimate of heading |
| $\tilde{m}_t^{comb}$ | estimated heading angle from Cue Combination model |
| $\tilde{m}_t^{hy}$ | estimated heading angle from Hybrid model |
| *Estimates of task/subject parameters* | |
| $s_A$ | subject's estimate of variability of the memory noise (i.e. their estimate of $\sigma_A$) |
| $r$ | subject's estimate of the prior probability that the feedback is true |
| $s_f$ | subject's estimate of variability of true feedback (i.e. subject's estimate of $\sigma_f$) |
| $p_{true}$ | subject's estimate that feedback on current trial is true (= $p(\text{true}|f, \delta_{1:t_f})$) |
| *Parameters related to target angle* | |
| $A$ | subject's remembered target angle |
| $\beta_A$ | subject's bias in remembered target angle |
| $\gamma_A$ | subject's gain in remembered target angle |
| $n_A$ | subject's noise in remembered target angle |
| $\sigma_A$ | standard deviation of noise in subject's remembered target angle |
| *Parameters related to velocity* | |
| $d$ | subject's estimate of velocity |
| $\gamma_d$ | gain in subject's estimate of velocity |
| $v$ | noise in subject's estimate of velocity |
| $\sigma_d$ | standard deviation of noise in subject's estimate of velocity |
| $s_d$ | subject's estimate of their own velocity noise |

that the participant's memory of the target angle is

$$A = \gamma_A \alpha + \beta_A + n_A \tag{4}$$

where $\gamma_A$ and $\beta_A$ are the gain and bias on the memory that leads to systematic over- or under-estimation of the target angle, and $n_A$ is zero mean Gaussian noise with variance $\sigma_A^2$.

To determine the response, we assume that participants stop moving when their current heading estimate matches the remembered angle. That is, when

$$m_t = A \tag{5}$$

Substituting in the expressions for $m_t$ and $A$ (from Section 1 of S1 File), we can then compute

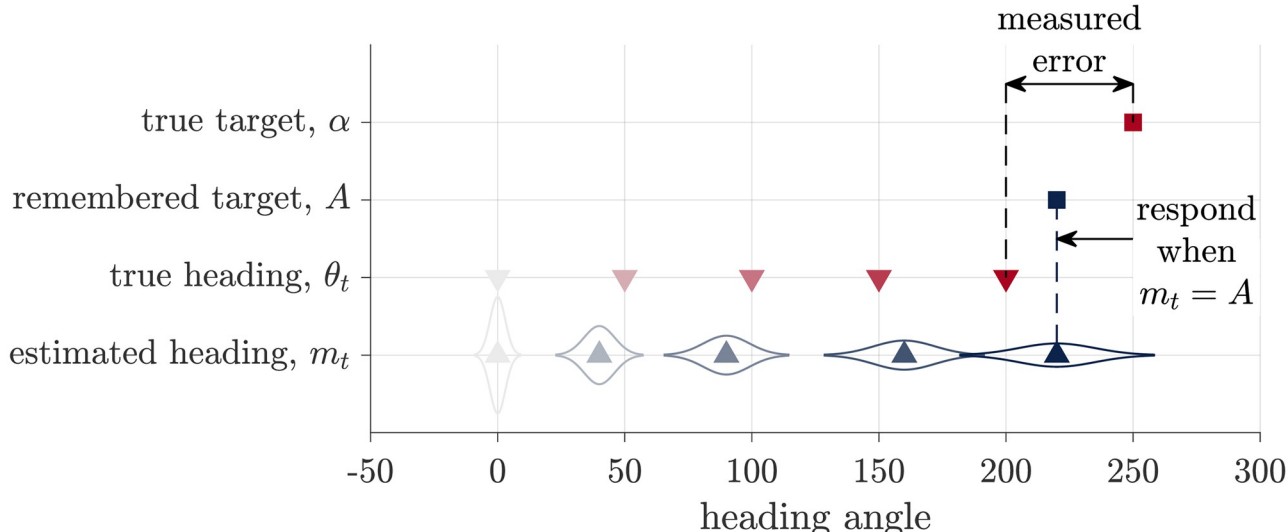

**Fig 2. Schematic of the Path Integration model.** During path integration, participants keep track of a probability distribution over their heading, which is centered at mean $m_t$. To respond they compare this estimated heading to their remembered target location, $A$, halting their turn when $m_t = A$. The experimenter observers neither of these variables, instead we quantify the measured error as the difference between the true target angle, $\alpha$, and the true heading angle, $\theta_t$.

the distribution over the measured error; i.e., the difference between participants actual heading and the target $(\theta_t - \alpha)$. Assuming all noises are Gaussian implies that the distribution over measured error is also Gaussian with a mean and variance given by

$$\mathbb{E}[\theta_t - \alpha] = \frac{(\gamma_A - \gamma_d)\alpha + \beta_A}{\gamma_d} \quad ; \quad \mathbb{V}[\theta_t - \alpha] = \frac{\sigma_d^2 \alpha + \sigma_A^2}{\gamma_d^2} \tag{6}$$

Thus, the Path Integration model predicts that both the mean error and the variance in the mean error will be linear in the target angle, $\alpha$, a prediction that we can test in the No Feedback condition. In addition, in the Feedback condition, the Path Integration model also predicts that the response error in the Feedback condition will be independent of the visual feedback (Fig 3A), a result that should not be surprising given that the Path Integration model ignores visual feedback.

**Kalman Filter model.** Unlike the Path Integration model, which always ignores feedback, the Kalman Filter model always incorporates the visual feedback into its estimate of heading (Fig 4). Thus the Kalman Filter model captures one of the key features of the Landmark Navigation strategy. However, it is important to note that the Kalman Filter model is slightly more general than 'pure' Landmark Navigation. Indeed, for most parameter values, it is a cue combination model in that it combines the the visual feedback with the estimate from Path Integration. Only for some parameter settings (as we shall see below), does the Kalman Filter model converge to a pure Landmark Navigation strategy in which it completely ignores prior idiothetic cues when visual feedback is presented.

The Kalman Filter model breaks down the retrieval phase of the task into four different stages: initial path integration, before the visual feedback is presented; feedback incorporation, when the feedback is presented; additional path integration, after the feedback is presented; and target comparison, to determine when to stop.

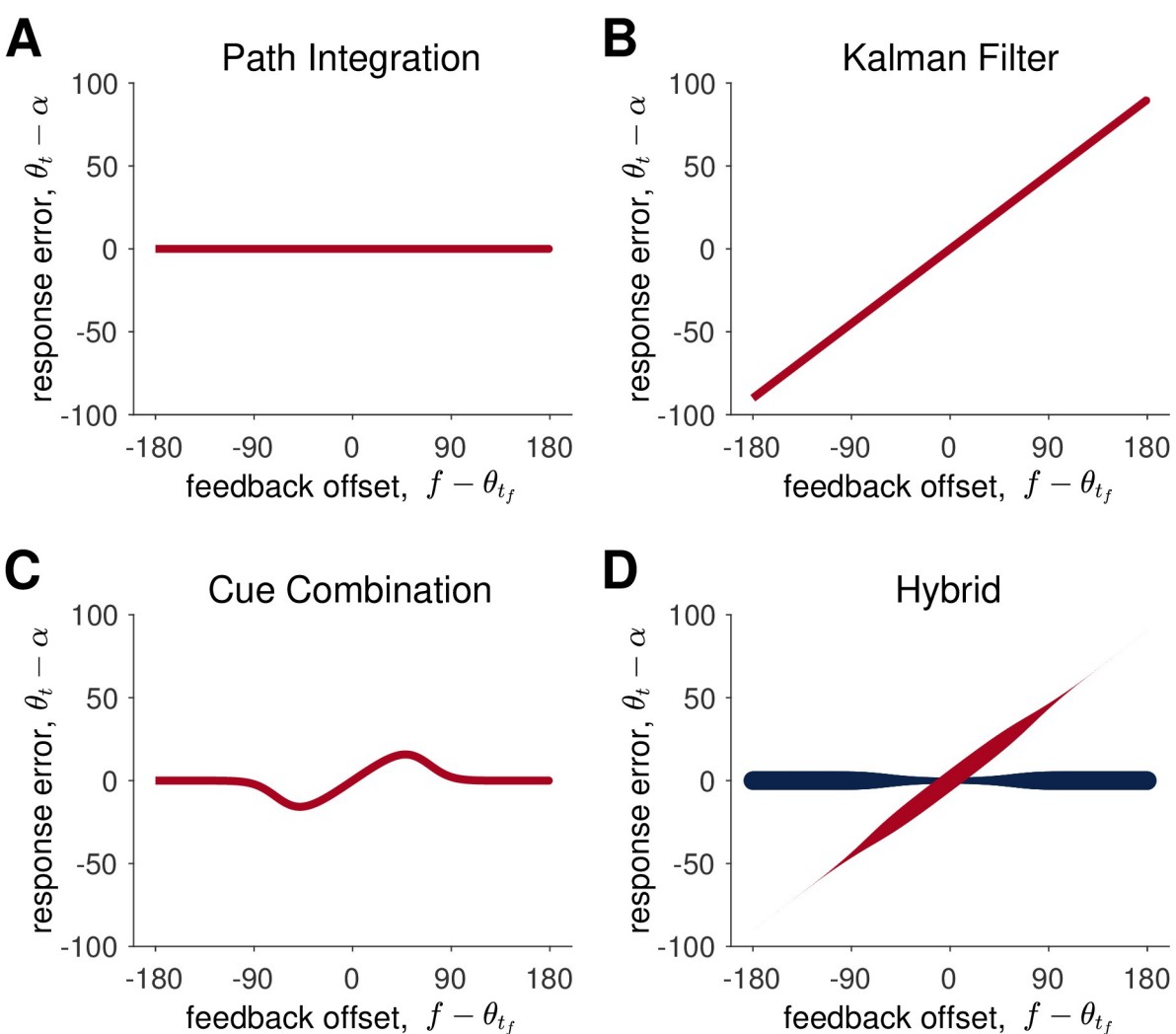

**Fig 3. Model predictions for the Path Integration, Kalman Filter, Cue Combination, and Hybrid models.** In (A-C) the red lines correspond to the mean of the response error predicted by the model. In (D) the two lines correspond to the mean response when the model assumes the feedback is true (red) and false (blue). The thickness of the red and blue lines in (D) corresponds to the probability that the model samples from a distribution with this mean, i.e. $p_{true}$ for red and $1 - p_{true} = p_{false}$ for blue.

Initial path integration, is identical to the Path Integration model (used in No Feedback trials). The model integrates noisy angular velocity information over time to form an estimate of the mean, $m_t$ and uncertainty, $s_t$, over the current heading angle $\theta_t$.

When feedback ($f$) is presented, the Kalman Filter model incorporates this feedback with the estimate from the initial path integration process in a Bayesian manner. Assuming all distributions are Gaussian, the Kalman Filter model computes the posterior distribution over head direction as

$$p(\theta_{t_f}|f, d_{1:t_f-1}) \quad = \mathcal{N}(\theta_{t_f}|\hat{m}_{t_f}, \hat{s}^2_{t_f}) \tag{7}$$

where $\hat{s}^2_{t_f}$ is the variance of the posterior (whose expression is given in the Section 1 of S1 File)

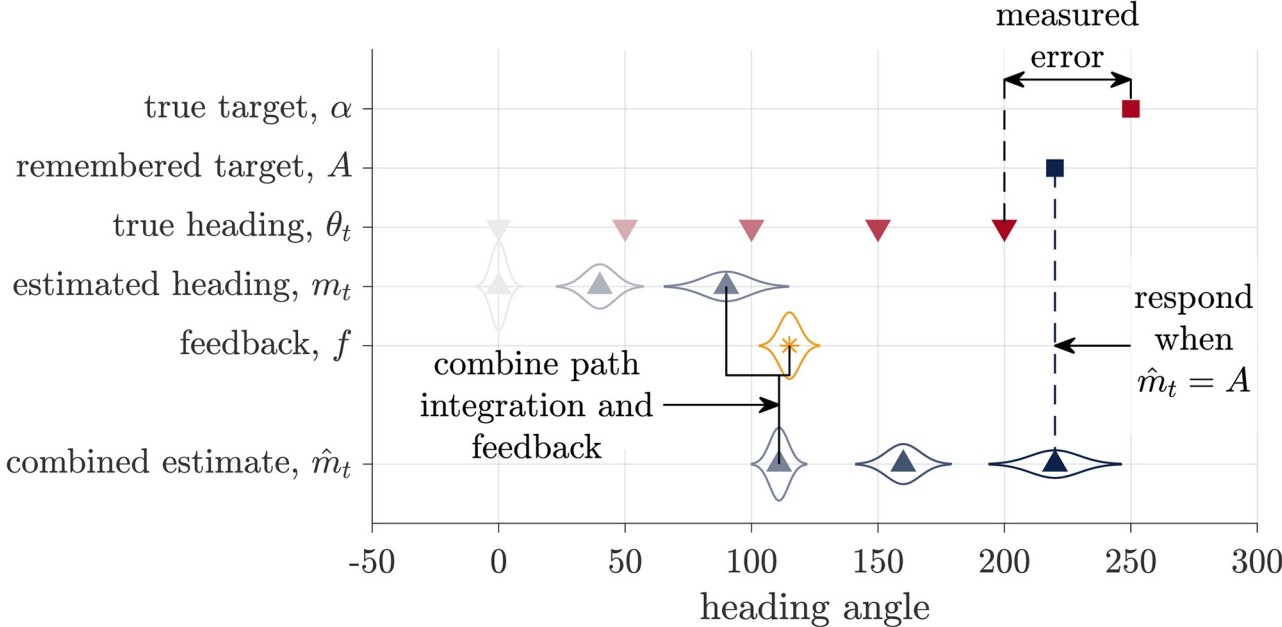

**Fig 4. Schematic of the Kalman Filter model.** Similar to the Path Integration model, this models assumes that participants keep track of a probability distribution over their heading that, before the feedback, is centered on mean $m_t$. When the feedback, $f$, is presented, they combine this visual information with their path integration estimate to compute a combined estimate of heading $\hat{m}_t$. They then stop turning and register their response when $\hat{m}_t = A$, their remembered target. As with the Path Integration model, none of these internal variables are observed by the experimenter, who instead measures the error as the difference between the true target, $\alpha$, and heading angle $\theta_t$.

and $\hat{m}_{t_f}$ is the mean of the posterior given by

$$\hat{m}_{t_f} = m_{t_f} + K_{t_f}(f - m_{t_f}) \tag{8}$$

where $K_{t_f} \in [0, 1]$ is the 'Kalman gain,' sometimes also called the learning rate [46, 47].

The Kalman gain is a critical variable in the Kalman Filter model because it captures the relative weighting of idiothetic (i.e. the estimate from Path Integration, $m_{t_f}$) and allothetic (i.e. the feedback, $f$) information in the estimate of heading. In general, the Kalman gain varies from person to person and from trial to trial depending on how reliable people believe the feedback to be relative to how reliable they believe their path integration estimate to be. When the model believes that the Path Integration estimate is more reliable, the Kalman gain is closer to 0 and idiothetic cues are more heavily weighted. When the model believes that the feedback is more reliable, the Kalman gain is closer to 1 and the allothetic feedback is more heavily weighted. In the extreme case that the model believes that the feedback is perfect, the Kalman gain is 1 and the Kalman Filter model implements 'pure' landmark navigation, basing its estimate of heading entirely on the visual feedback and ignoring the path integration estimate completely.

After the feedback has been incorporated, the model continues path integration using noisy velocity information until its estimate of heading matches the remembered target angle. Working through the algebra (see Section 1 of S1 File) reveals that the measured response distribution is Gaussian with a mean given by

$$\mathbb{E}[\theta_t - \alpha] \quad = \frac{1}{\gamma_d}\Big((\gamma_A - \gamma_d)\alpha - K_{t_f}(f - \gamma_d\theta_{t_f}) + b\Big) \tag{9}$$

 

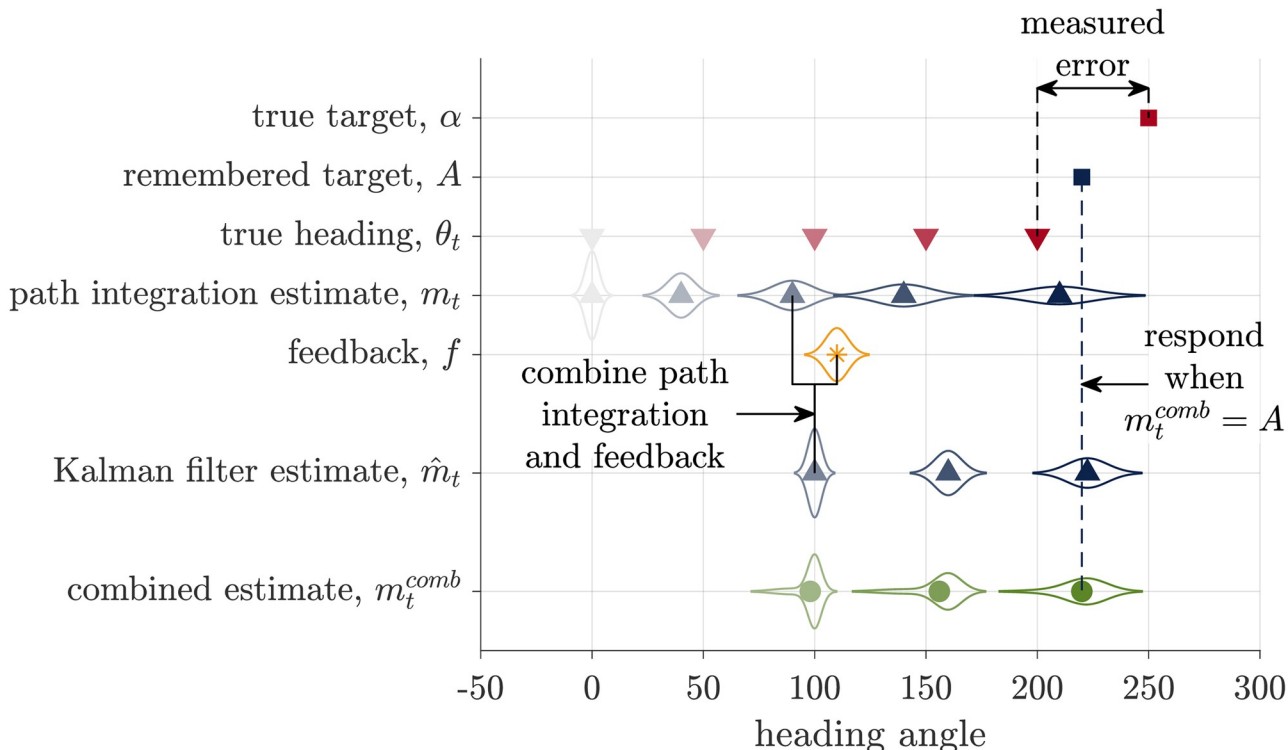

**Fig 5. Schematic of the Cue Combination model.** The Cue Combination model combines two estimates of heading, the path integration estimate, $m_t$, and the Kalman Filter estimate, $\hat{m}_t$, to compute a combined estimate $m_t^{comb}$. The response is made when this combined estimate matches the remembered target $A$.

This implies that the error in the Kalman Filter model is linear in the feedback prediction error $f - \gamma_d \theta_{t_f}$ (Fig 3B).

**Cue Combination model.** Like the Kalman Filter model, the Cue Combination model combines the feedback with the estimate of heading from path integration. Unlike the Kalman Filter model, however, the Cue Combination model also takes into account the possibility that the feedback will be misleading, in which case the influence of the feedback is reduced. In particular, the Cue Combination model computes a mixture distribution over heading angle with one component of the mixture assuming that the feedback is false and the other that the feedback is true. These two components are weighed according to the computed probability that the feedback is true, $p_{true}$ (Fig 5).

Mathematically, the Cue Combination model computes the probability distribution over heading angle by marginalizing over the truth of the feedback

$$p(\theta_{t_f} | f, d_{1:t_f}) = p(\theta_{t_f} | \text{false}, d_{1:t_f}) p_{false} + p(\theta_{t_f} | \text{true}, f, d_{1:t_f}) p_{true} \tag{10}$$

where $p_{true} = 1 - p_{false} = p(\text{true} | f, d_{1:t_f})$ is the probability that the feedback is true given the noisy velocity cues seen so far. Consistent with intuition, $p_{true}$ decreases with the absolute value of the prediction error at the time of feedback ($f - m_{t_f}$) such that large prediction errors are deemed unlikely to come from true feedback.

Eq 10 implies that, at the time of feedback, the Cue Combination model updates its estimate of the mean heading by <u>combining</u> the estimates from Path Integration model, $m_{t_f}$, with the

 

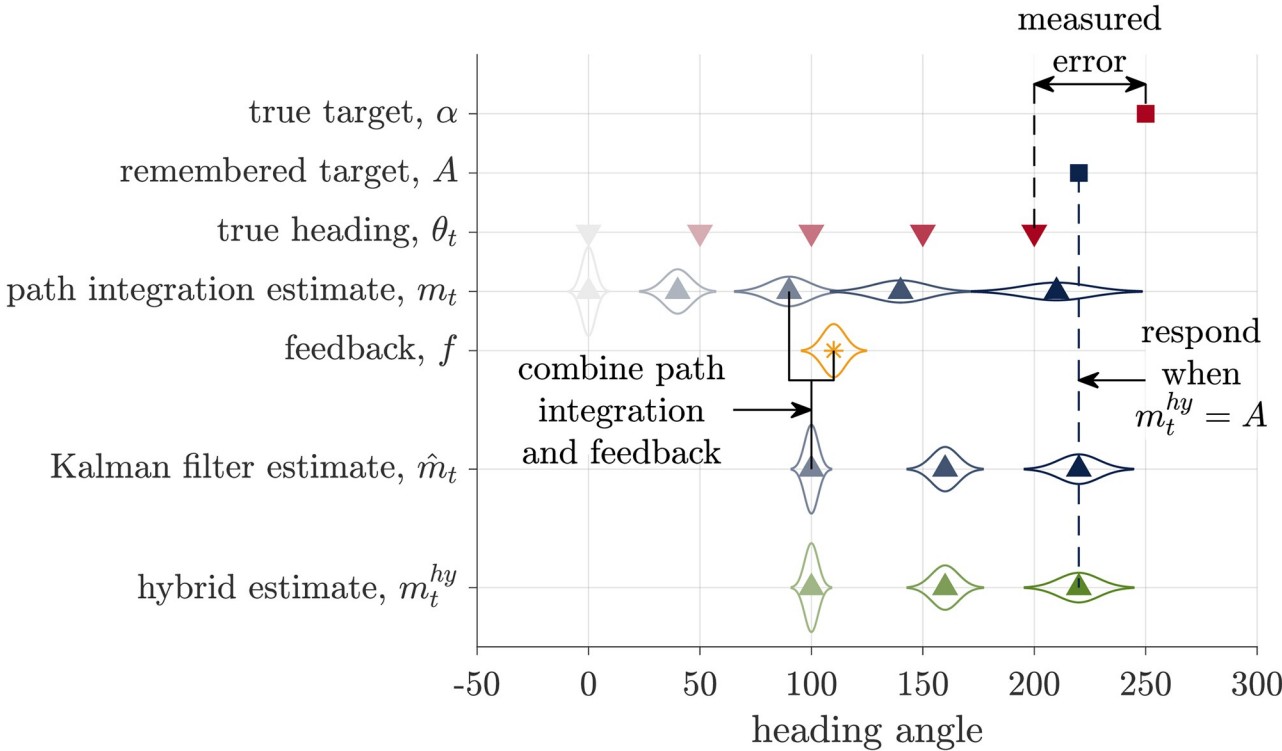

**Fig 6. Schematic of the Hybrid model.** The Hybrid model bases its estimate of heading, $m_t^{hy}$, either on the path integration estimate or the Kalman Filter estimate. Here we illustrate the case where the model chooses the Kalman Filter estimate. The response is made when the hybrid estimate matches the remembered target angle.

estimate from the Kalman Filter model, $\hat{m}_{t_f}$, as

$$\tilde{m}_{t_f}^{comb} = m_{t_f} p_{false} + \hat{m}_{t_f} p_{true} \tag{11}$$

Assuming that a similar target comparison process determines the response (i.e. participants stop turning when $\tilde{m}_t^{comb} = A$), then this implies that the response distribution for the Cue Combination model will be Gaussian with a mean error given by the mixture of the Path Integration and Kalman Filter responses:

$$\mathbb{E}[\theta_t - \alpha] = \frac{1}{\gamma_d} \left( (\gamma_A - \gamma_d)\alpha - K_{t_f} p_{true} (f - \gamma_d \theta_{t_f}) + \beta_A \right) \tag{12}$$

Because $p_{true}$ depends on the prediction error, Eq 12 implies that the average error in the Cue Combination has a non-linear dependence on the prediction error (Fig 3C).

**Hybrid model.** Instead of averaging over the possibility that the feedback is true or false, in the Hybrid model the estimates from the Path Integration model and the Kalman Filter model compete (Fig 6).

In particular, we assume that the Hybrid model makes the decision between Path Integration and Kalman Filter estimates according to the probability that the feedback is true ($p_{true}$), by underline{sampling} from the distribution over the veracity of the feedback. Thus with probability $p_{true}$, this model behaves exactly like the Kalman Filter model, setting its estimate of heading to $m_t^{hy} = \tilde{m}_t$, and with probability $p_{false} = 1 - p_{true}$ this model behaves exactly like the Path Integration model, setting its estimate of heading to $m_t^{hy} = m_t$. This implies that the distribution of

errors is a mixture of the Kalman Filter and Path Integration models such that the average response error is

$$
\mathbb{E}(\theta_t - \alpha) = \begin{cases} \dfrac{1}{\gamma_d}\left((\gamma_A - \gamma_d)\alpha - K_{t_f}(f - \gamma_d\theta_{t_f}) + b\right) & \text{with probability } p_{true} \\[12pt] \dfrac{1}{\gamma_d}((\gamma_A - \gamma_d)\alpha + \beta_A) & \text{with probability } 1 - p_{true} \end{cases}
\tag{13}
$$

This competition process ensures that the relationship between response error and feedback offset will be a mixture of the Path Integration and Kalman Filter responses (Fig 3D). When the model decides to ignore the feedback, the response will match the Path Integration model. This occurs most often for large offset angles, when $p_{true}$ is closer to 0. When the model decides to incorporate the feedback, the response will lie on the red line. This occurs most often for small offset angles, when $p_{true}$ is closer to 1.

**Summary of models.** The behavior of the models on the feedback trials is summarized in Fig 3. In addition, a summary of the parameters in each model is shown in Table 2.

The Path Integration model ignores all incoming feedback, basing its estimate of location entirely on the estimate from the path integration of idiothetic cues. Thus, in the feedback condition, the response error is independent of the feedback offset (Fig 3A). The Path Integration model technically has five free parameters. However, because these parameters all appear as ratios with $\gamma_d$, only the four ratios ($\sigma_d/\gamma_d$ etc . . .) can be extracted from the fitting procedure (Table 2).

The Kalman Filter model always integrates visual information regardless of the offset angle. Thus, the response error grows linearly with the feedback offset (Fig 3B). The Kalman Filter model has eight free parameters, three of which appear as ratios such that only seven free parameters can be extracted from the data (Table 2).

The Cue Combination model combines the estimates from the Path Integration model with the Kalman Filter model according to the probability that it believes the feedback is true. This

**Table 2. Parameters, their ranges and values, in the different models.** The presence of a parameter in a model is indicated by either a check mark (when it can be fit on its own), a ratio (when it can be fit as a ratio with another parameter), or = 1 when it takes the value 1.

| Parameter and range | Path Integration | Kalman Filter | Cue Combination | Hybrid |
|---|---|---|---|---|
| Velocity gain, $\gamma_d$ <br> $0 \leq \gamma_d \leq 4$ | = 1 | ✓ | ✓ | ✓ |
| Variance of velocity noise, $\sigma_d^2$ <br> $0 \leq \sigma_d \leq 20$ | $\sigma_d/\gamma_d$ | ✓ | ✓ | ✓ |
| Target gain, $\gamma_A$ <br> $0 \leq \gamma_A \leq 2$ | $\gamma_A/\gamma_d$ | ✓ | ✓ | ✓ |
| Target bias, $\beta_A$ <br> $-180 \leq \beta_A \leq +180$ | $\beta_A/\gamma_d$ | ✓ | ✓ | ✓ |
| Variance of target noise, $\sigma_A^2$ <br> $0 \leq \sigma_A \leq 20$ | $\sigma_A/\gamma_d$ | ✓ | ✓ | ✓ |
| Participant's initial uncertainty, $s_0^2$ <br> $0 \leq s_0 \leq 20$ | | $s_0/s_f$ | ✓ | ✓ |
| Participant's velocity noise variance, $s_d^2$ <br> $0 \leq s_d \leq 20$ | | $s_d/s_f$ | ✓ | ✓ |
| Participant's feedback noise variance, $s_f^2$ <br> $0 \leq s_f \leq 50$ | | = 1 | ✓ | ✓ |
| Participant's prior on true feedback, $r$ <br> $0 \leq r \leq 1$ | | | ✓ | ✓ |

leads to a non-linear relationship between feedback offset and response error (Fig 3C). This model has nine free parameters, all of which can be extracted from the fitting procedure (Table 2).

The Hybrid model also uses the estimate from Path Integration model and the Kalman Filter model. However, instead of combining them, it chooses one or the other depending on the probability with which it believes the feedback is true. This process leads to bimodal responses from large feedback offsets (Fig 3D). This model also has nine free parameters, all of which can be extracted from the fitting procedure (Table 2).

## Model fitting and comparison

Each model provides a closed form function for the likelihood that the a particular angular error is observed on each trial, $\tau$, given the target, the feedback (in the Feedback condition), and the true heading angle at feedback. That is, we can formally write the likelihood of observing error$^\tau$ on trial $\tau$ as

$$p(\text{error}^\tau | \mathbf{X}) = \begin{cases} p(\text{error}^\tau | \alpha^\tau, \mathbf{X}) & \text{No Feedback condition} \\ p(\text{error}^\tau | \alpha^\tau, f^\tau, \theta_{t_f}^\tau, \mathbf{X}) & \text{Feedback condition} \end{cases} \tag{14}$$

where vector $\mathbf{X}$ denotes the free parameters of the model. In all cases we used the Path Integration model to compute the likelihoods on the No Feedback trials and each of the four models to compute the likelihoods on the Feedback trials. When combining each model with the Path Integration model in this way, we yoked the shared parameters between the models to be equal across the No Feedback and Feedback trials.

We then combined the likelihoods across trials to form the log likelihood for a given set of parameters

$$LL(\mathbf{X}) = \sum_\tau \log p(\text{error}^\tau | \mathbf{X}) \tag{15}$$

where the sum is over the trials in both the No Feedback and Feedback conditions. The best fitting parameters were then computed as those that maximize this log likelihood

$$\mathbf{X}_{MLE} = \underset{\mathbf{X}}{\text{argmax}}\, LL(\mathbf{X}) \tag{16}$$

Model fitting was performed using the fmincon function in Matlab. To reduce the possibility of this optimization procedure getting trapped in local minima, we ran this process 100 times using random starting points. Each starting point was randomly sampled between the upper and lower bound on each parameter value as defined in Table 2. Parameter recovery with simulated data showed that this procedure was able to recover parameters adequately for all models (Section 3 of S1 File and S6 and S8–S10 Figs).

Model comparison was performed by computing the Bayes Information Criterion (BIC) for each model for each participant

$$BIC = k \log n - 2LL(\mathbf{X}_{MLE}) \tag{17}$$

where $k$ is the number of free parameters in the model and $n$ is the number of trials in the data. Model recovery with simulated data showed that this procedure was sufficient to distinguish between the four models on this experiment (Section 3 of S1 File and S4 Fig).

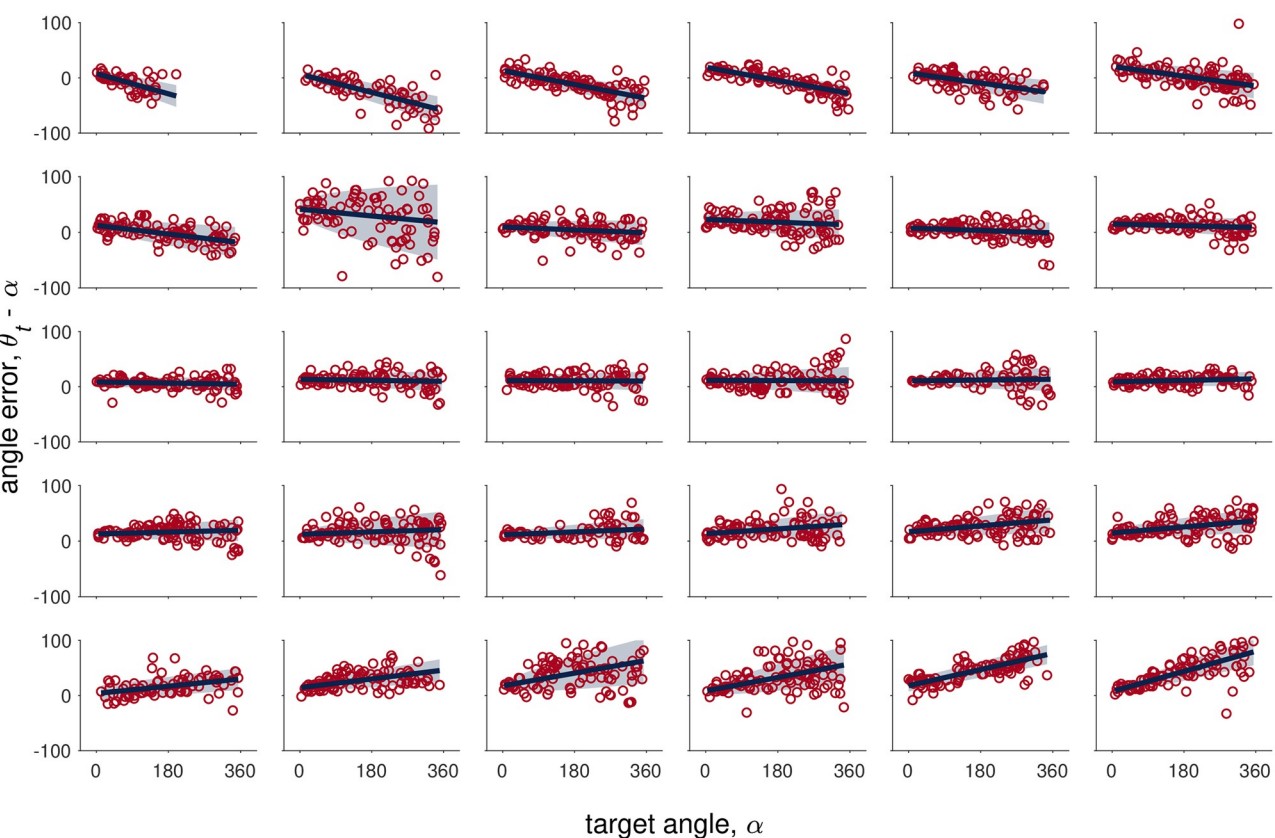

**Fig 7. Error vs target angle for the No Feedback condition.** Each plot corresponds to data from one participant and plots are ordered from most negative slope (top left) to most positive slope (bottom right). The red circles correspond to human data, the solid blue to the mean error from the Path Integration model fit, and shaded blue area to the mean ± standard deviation of the error from the Path Integration model fit.

## Results

### Behavior in the No Feedback condition is consistent with the Path Integration model

The Path Integration model predicts that the mean of the response error will be linear in the target angle $\alpha$. To test whether this linear relationship holds, we plotted the response error $\theta_t - A$ as a function of target angle $\alpha$ for all of the No Feedback trials (Fig 7). This reveals a clear linear relationship between the mean response error and target angle. In addition, for many participants, the variability of the response error also appears to increase with the target angle, which is also consistent with the Path Integration model. Notable in Fig 7 are the considerable individual differences between participants, with some participants having a negative slope (systematically underestimating large target angles), some a positive slope (overestimating large target angles), and some with approximately zero slope.

To investigate further, we fit the Path Integration model to the No Feedback data. This model has five free parameters capturing the gain and noise in the velocity signal ($\gamma_d$, $\sigma_d$) and the gain, bias and noise in the target encoding process ($\gamma_A$, $\beta_A$, $\sigma_A$). Because $\gamma_d$ only appears as part of a ratio with other parameters in the Path Integration model, it cannot be estimated separately. We therefore fix the value of the velocity gain to $\gamma_d = 1$ and interpret the resulting parameter values as ratios (e.g. $\gamma_A/\gamma_d$ etc . . .).

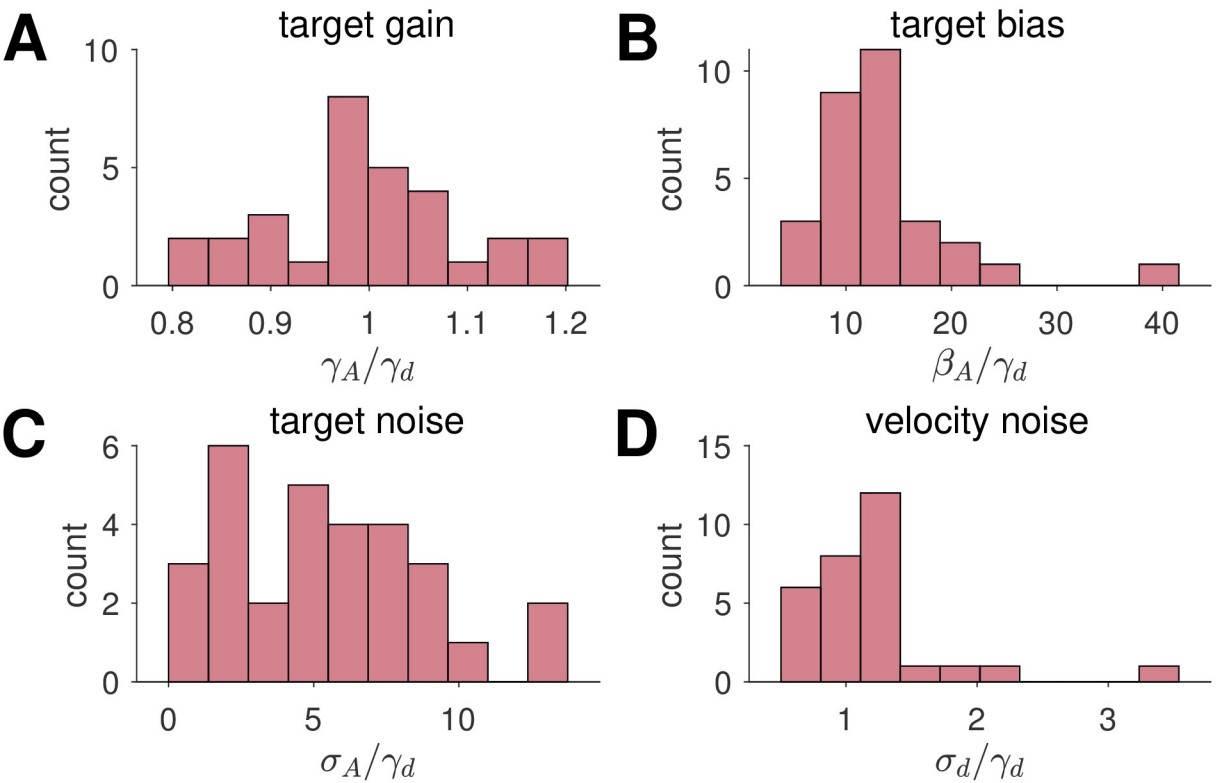

**Fig 8. Parameter values for the Path Integration model fit to the No Feedback data.** Histograms show the distribution parameter values across participants. The counts are the number of participants, whose fitted parameter values fall within each bin.

As shown in Fig 7, the Path Integration model provides an excellent fit to the No Feedback data, accounting for both the linear relationship between response error and target and the increase in variability with target.

Looking at the best fitting parameter values, we find that the target gain is close to 1 at the group level (mean $\gamma_A/\gamma_d$ = 0.997), indicating no systematic over- or under-weighting of the target across the population (Fig 8A). Individual participants vary considerably, however, with $\gamma_A/\gamma_d$ ranging from 0.8 (negative slope in Fig 7) to 1.2 (positive slope in Fig 7). In contrast to the target gain, we find a systematic target bias across the population, with all participants turning slightly too far (mean $\beta_A/\gamma_d$ = 13.4˚; Fig 8B). Nonetheless, as can be seen in Fig 7, there is considerable variability across participants.

For accuracy, we find that most participants have some target noise (mean $\sigma_A/\gamma_d$ = 5.35˚; Fig 8C) and all participants have velocity noise (mean $\sigma_d/\gamma_d$ = 1.19˚; Fig 8D). This latter result suggests that the variance of the noise in head direction estimates grows linearly with target angle and with a constant of proportionality close to 1.

## Behavior in the Feedback condition is consistent with the Hybrid model

The key analysis for the Feedback condition relates the feedback offset, $f - \theta_{t_f}$, to the response error, $\theta_t - \alpha$. As illustrated in Fig 3, each model predicts a different relationship between these variables.

In our experiment we found examples of behavior that was qualitatively consistent with all four models. These are illustrated in Fig 9. At the extremes, Participant 27 appeared to ignore

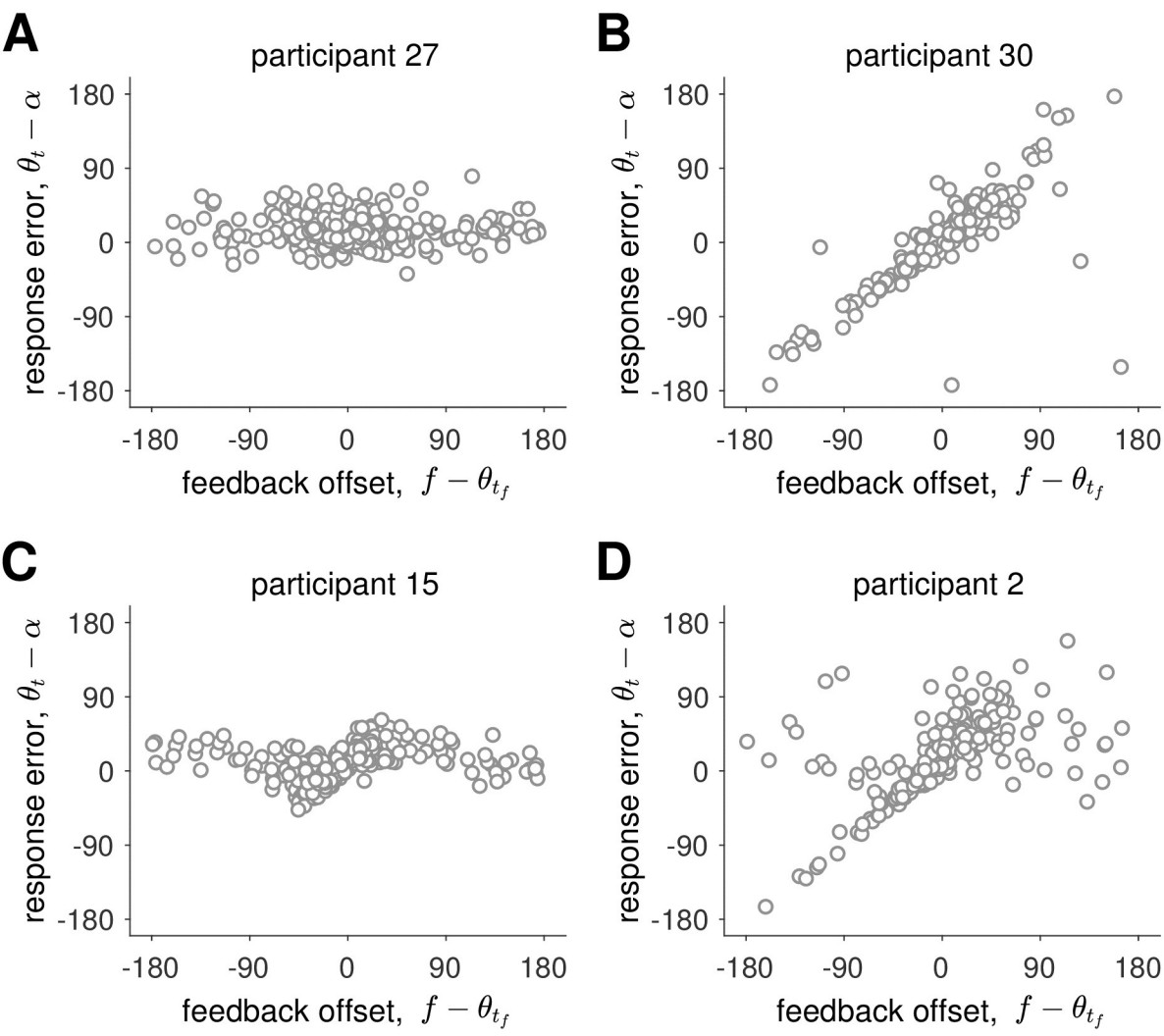

**Fig 9. Examples of human behavior on the feedback trials.**

feedback completely, similar to the Path Integration model (Fig 9A), while Participant 30 seemed to always use the feedback, just like the Kalman Filter model (Fig 9B). Conversely, Participant 15 appeared to use a Cue Combination approach, while Participant 2's behavior was more consistent with the Hybrid model. This latter behavior is especially interesting because it strongly suggests a bimodal response distribution for large feedback offsets.

To quantitatively determine which of the four models best described each participant's behavior we turned to model fitting and model comparison. We computed Bayes Information Criterion (BIC) scores for each of the models for each of the participants, which penalizes the likelihood values of each model fit by the number of free parameters. The Hybrid and Cue Combination models have 9 free parameters, while the Kalman filter and Path integration models have 8 and 5 free parameters respectively (Table 2). Fig 10A plots BIC scores for each model relative to the BIC score for the Hybrid model for each participant. In this plot, positive values correspond to evidence in favor of the Hybrid model, negative values correspond to evidence in favor of the other models. As can be seen in Fig 10, the the Hybrid model is heavily favored and best describes the behavior of all but three participants (participant 27, who is best

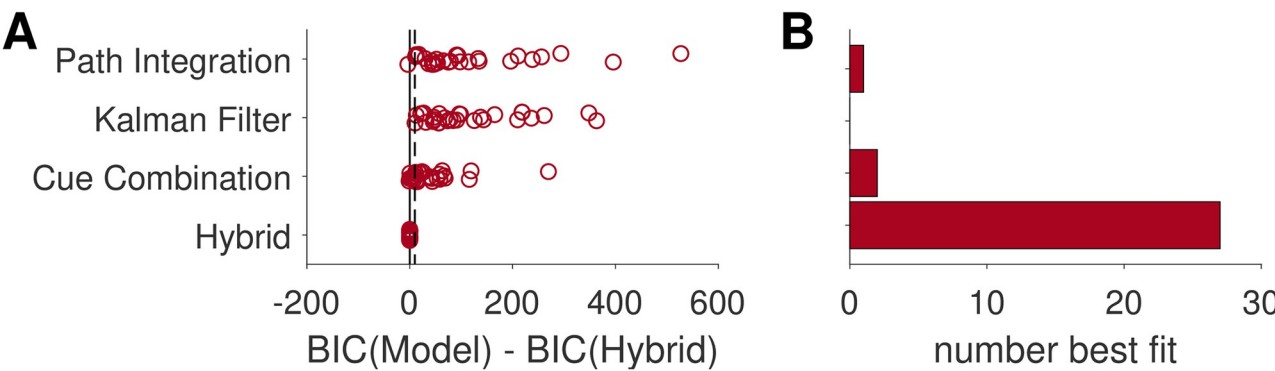

**Fig 10. Model comparison.** (A) BIC scores for each model relative to the BIC score for the Hybrid model for each participant. For each model, each circle corresponds to one participant. Positive numbers imply the fit favors the Hybrid model, negative numbers imply that the fit favors the other model. A ΔBIC value of > 10 (indicated by the dotted line) is considered "very strong" evidence implied by very high posterior odds ($P(M_{Hybrid} \mid Data)$ > 0.99) [48]. (B) The number of participants best fit by each model. 28 out of 30 participants were best fit by the Hybrid model, suggesting that this model best describes human behavior.

fit by the Path Integration model, and participants 15 and 23, who are best fit by the Cue Combination model; Fig 10B).

Qualitatively, the Hybrid model provides a good account of the data despite the large individual differences in behavior. In Fig 11 we compare the behavior of the model to the behavior of four example participants. As already suggested in Fig 9, Participant 2 is one of the cleanest examples of Hybrid behavior and it is not surprising that this behavior is well described by the model. Likewise the Hybrid model does an excellent job capturing the behavior of Participant 30, whose qualitative behavior appears more Kalman Filter like. The reason the Hybrid model outperforms the Kalman Filter model for this participant is that Participant 30 appears to ignore the stimulus on two trials at offsets of around -100 and +100 degrees. These data points correspond to large deviations from the Kalman Filter model behavior but are a natural consequence of the Hybrid model. The Hybrid model also captures the behavior of participants who integrate the feedback over a much smaller range such as Participants 10 and 25. A comparison between the Hybrid model and all participants is shown in S13 Fig.

### Parameters of the Hybrid model suggest people use a true hybrid strategy between cue combination and cue competition

Consistent with the individual differences in behavior, there were significant individual differences in the fit parameter values across the group (S11 Fig). Of particular interest is what these parameter values imply for the values of the Kalman gain, $K_{t_f}$. As mentioned in the Methods section, this variable is important because it determines the extent to which the Kalman Filter component of the Hybrid model incorporates the allothetic visual feedback vs the idiothetic path integration estimate of heading. The larger the Kalman gain, the more allothetic information is favored over idiothetic information. Moreover, if the Kalman gain is 1, then the Hybrid model becomes a 'pure' cue competition model. This is because the Kalman Filter component of the model ignores idiothetic information prior to the feedback (i.e. it implements 'pure' landmark navigation). Thus, the Hybrid model now decides between pure Path Integration and pure Landmark Navigation, consistent with a pure Cue Combination approach.

In Fig 12, we plot implied Kalman gains for all trials for each participant. This clearly shows that the majority of participants do not have $K_{t_f} = 1$, instead showing intermediate values for

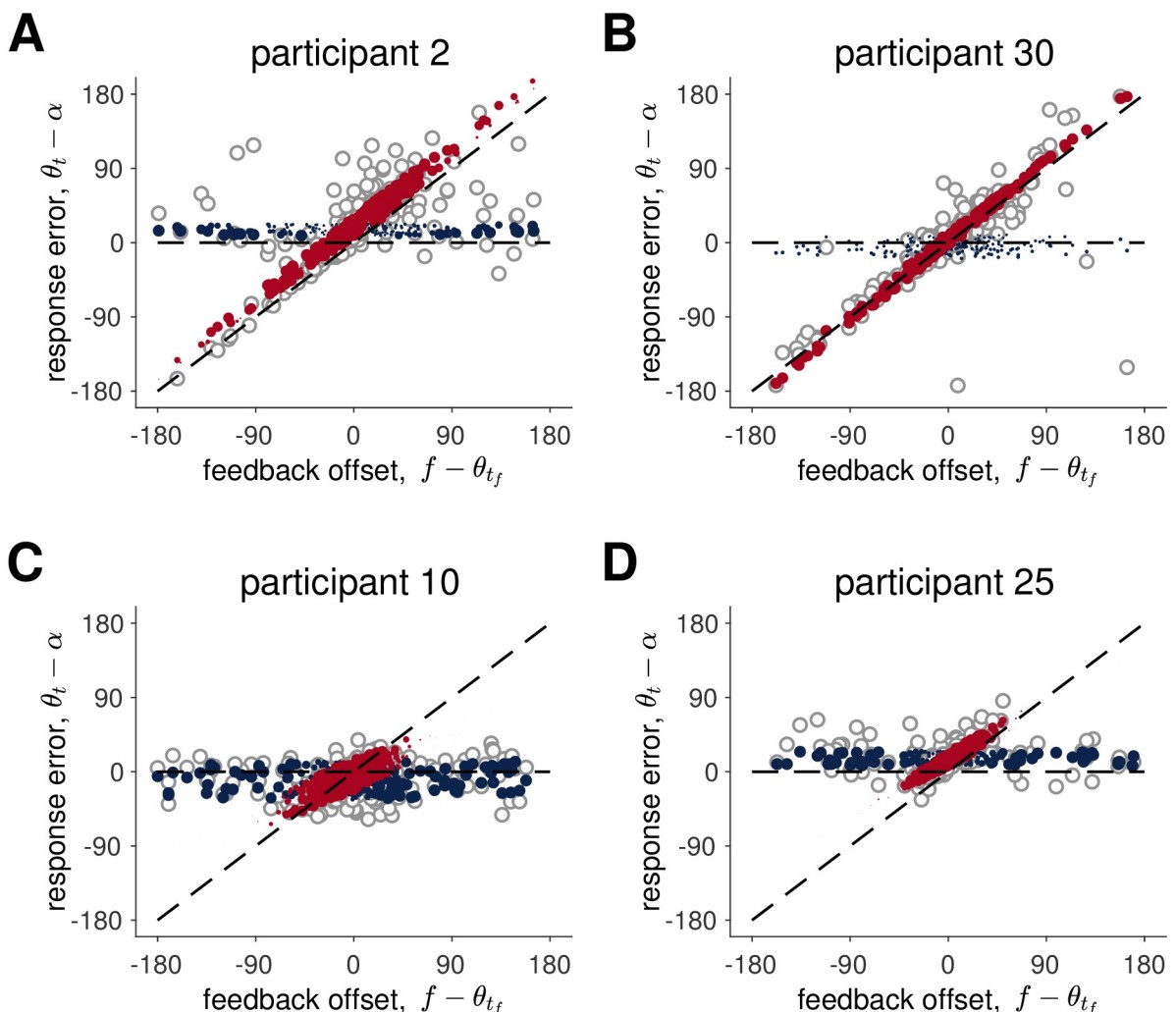

**Fig 11. Comparison between data and the Hybrid model for four participants.** Four participants' data (open grey dots) are overlaid hybrid model's mean responses when the model assumes the feedback is true (red) and false (blue). The size of the dots corresponds to the probability that the model samples from a distribution with this mean, i.e. $p_{true}$ for red and $1 − p_{true} = p_{false}$ for blue.

the Kalman gain. Thus we conclude that participants use a true hybrid of cue combination, when the mismatch between idiothetic and allothetic information is small, and cue competition when the mismatch is large.

## Discussion

In this paper, we investigated how humans integrate path integration and visual landmarks/boundaries to estimate their heading. In our experiment, the 'Rotation Task', participants made a series of turns in virtual reality, mostly without visual feedback. Visual feedback, when it was presented (in the form of the boundaries of the room with proximal landmarks), was brief and offset from the true heading angle. This offset led to systematic errors in people's turning behavior that allowed us to quantify how people combine visual allothetic feedback with their internal estimate of heading direction, computed by path integration of body-based idiothetic cues.

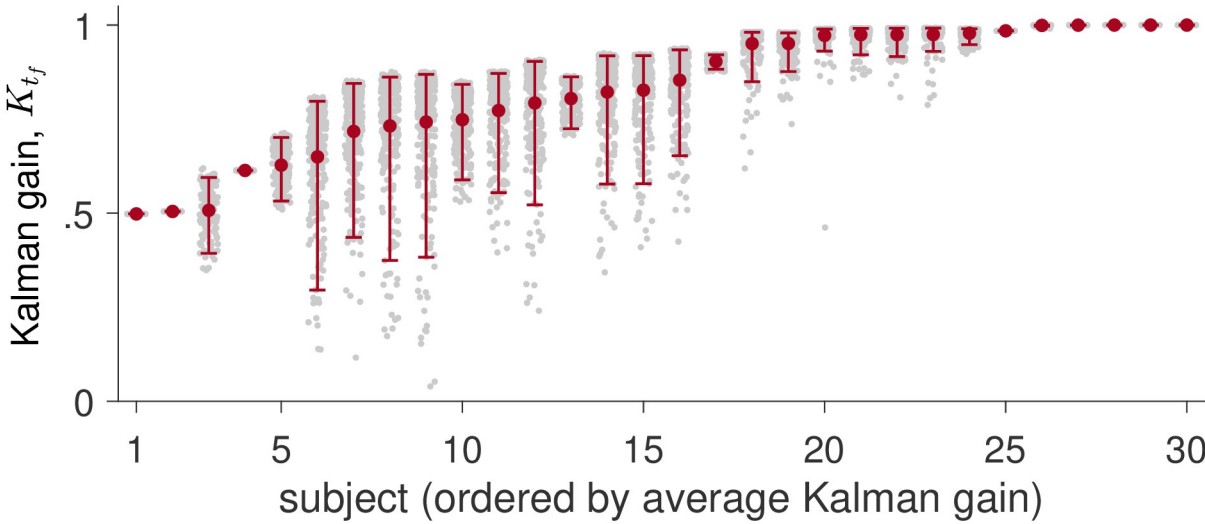

**Fig 12. Computed Kalman gain for all participants and all trials.** The Kalman gains computed for each trial for each participant are shown as gray dots. The mean Kalman gain and 95% confidence intervals are shown in red.

While there were considerable individual differences in task performance, our findings suggest that the majority of participants used the same overarching <u>hybrid</u> strategy to complete the task. In this strategy, body-based idiothetic and visual allothetic cues are combined when the estimates of path integration and landmark navigation are close and compete when the estimates are far apart. This behavior was well accounted for by a computational model that switches between competition and combination according to the subjective probability that the feedback is valid.

One limitation of this work is that we have focused only on rotation, ignoring translation completely. While this approach has the advantage of simplifying both the analysis and the task (e.g. removing the risk of participants accidentally walking into a real-world wall during their virtual navigation), it may be that we are missing something relevent to navigation, translations. Indeed, related to this point, Mou and colleagues [30, 31, 49] have argued that estimates from path integration and visual landmarks are combined differently depending on whether the task is self-localization (position and head direction) or homing (returning to the start position). Thus, while in the rest of the Discussion, we focus on the general implications of our work, a key for future work will be to expand our experimental paradigm and model to account for translational as well as rotational movements.

Our findings in support of a Hybrid model may help to explain the mixed reports in the literature regarding cue combination. Specifically, some studies report evidence of cue combination [23, 25, 29, 49] while others find evidence for others cue competition [24, 50]. One set of studies using a similar experimental paradigm involves the short-range homing task of Nardini and colleagues and shows evidence for both cue combination and competition depending on the conditions tested [25]. In this task, participants picked up three successive objects in a triangular path and returned them after a delay. During the return phase of the task, the experimenters manipulated the visual feedback to induce a 15 degree mismatch with the body-based cues. When both visual and body-based cues were present, Nardini et al. found that the variance of the response was smaller than when navigation relied on only one cue, consistent with the combination of visual and body-based cues in a Bayesian manner. However, when Zhao and Warren [24] increased the offset from 15 to 135 degrees, they found that participants

based their estimate of location either entirely on the visual cues (when the offset was small) or entirely on the body-based cues (when the offset was large), taking this as evidence for a cue-competition strategy. Thus, in the same task, participants appeared to switch from visual landmarks to path integration as the offset grew larger. Such behavior is consistent with Hybrid model, albeit with a Kalman gain that is equal to 1, which is slightly different to what we observed in our experiment (Fig 12).

One explanation for this difference between our results and Zhao and Warren's could be the amount of time that feedback was presented for. In [24], the offset feedback was presented continuously, whereas in our task the feedback was presented for only 300ms. Thus, participants in Zhao and Warren's experiment may have been more confident that the visual feedback was correct which, by Eq S13, would lead to a Kalman gain close to 1. Consistent with this idea, in a different study, Zhao and Warren [51] found, using a catch trials design, that visual landmark reliability increased with environmental stability. In addition, they observed individual differences in cue switching behavior with most individuals showing no cue switching behavior at all. This suggests that for continuous stable visual feedback the Kalman gain will approach 1 for most participants. Interestingly our visual feedback was not continuous and was only moderately stable, yet several participants had a Kalman gain close to 1 (Eq S13). Given these results, an increased visual feedback duration would likely result in more reliance on visual cues and hence a general increase in the Kalman gain. A critical question for future work will be to ask how the Kalman gain changes as a function of viewing duration and a range of different environmental stabilities.

More generally, our model fits with bounded rationality theories of human cognition [52–55]. That is, people have limited computational resources, which in turn impacts the kinds of computations they can perform and how they perform them. In our case, combining visual and body-based cues to compute a probability distribution over heading should be easier when the cues align and the distribution is unimodal than when the cues conflict and the posterior is bimodal. In this latter case, representing the bimodal distribution by sampling one mode or the other, as the Hybrid model does, may be a rational strategy that demands fewer computational resources. Indeed, other work has shown that people may represent complex and multimodal distributions with a small number of samples, which may be as low as just a single sample in some cases [56–58].

A key prediction of such a sampling interpretation of the Hybrid model is that participants should sometimes lose information when integrating visual allothetic and body-based idiothetic cues. Similar to Robust cue integration [59], when faced with a large feedback offset, instead of computing the full posterior distribution over heading, participants collapse this bimodal distribution to a unimodal distribution centered on the estimate from path integration or landmark navigation, ignoring the less reliable cue.

Such a '*semi*-Bayesian' or interpretation, stands in contrast to a fully-Bayesian [21, 59–61] alternative models studied in visual perception in which, participants do indeed keep track of the bimodal or mixture posterior and instead sample their estimate of heading direction from this posterior to determine their response. In this view, when faced with a large feedback offset, participants do compute the full distribution over heading, but rather than average over this distribution to compute their response, they sample from it instead. This implies that participants do not make a decision to ignore or incorporate the feedback and, as a result, do not lose information about the stimulus or their path integration estimate.

A key question for future work will be to distinguish between these two interpretations of the task. Does sampling occur at the time of feedback causing a collapse of the posterior distribution to one mode and a loss of information? Or does sampling occur later on and without the collapse of the posterior? Both interpretations lead to identical behavior on the Rotation

Task. However, a modified version of the task should be able to distinguish them. One way to do this would be with two turns and two sets of visual feedback rather than one. In this task, participants would turn first to landmark A (e.g. the door) and then to landmark B (e.g. the window). The turn to landmark A would be identical to the task in this paper, with a brief flash of offset feedback along the way. After reporting their estimate of the location of landmark A, and crucially without feedback as to whether they were correct, participants would then make a second turn to landmark B. This second turn would also include a flash of visual feedback. If this second flash aligned with one mode of the bimodal posterior it should reinforce that mode if participants kept track of it. However, if they collapsed their posterior to a single mode, a flash at the other mode would have less effect. Thus, in principle, the two accounts could be distinguished.

Future work should also combine the Rotation Task with physiological measures to study the neural underpinnings of this process. Previous computational models based on line attractor neural networks produce behavior that is almost identical to the Hybrid model, combining cues when they are close together and picking one when they are far apart [62–67]. Moreover, recent findings in fruit flies suggest that, at least in insects, such networks may actually be present in the brain [68]. Investigating the link between the Hybrid model and its neural implementation should be a fruitful line of research.

Finally, it will be interesting to explore individual differences in behavior on the Rotation Task in more diverse populations including children, older adults, and people with psychiatric disorders. By providing within-trial dynamics of cognitive variables as well as characterizing large individual differences with different parameter values, our task and model could help to set the stage for this future work.

## Supporting information

**S1 Fig. Graphical representation of the Path Integration model.**
(TIF)

**S2 Fig. Graphical representation of the Kalman Filter model.**
(TIF)

**S3 Fig. Graphical representation of the Cue Combination and Hybrid models.**
(TIF)

**S4 Fig. Model recovery confusion matrix.**
(TIF)

**S5 Fig. Parameter recovery for Path Integration model.**
(TIF)

**S6 Fig. Parameter recovery for Hybrid mode.**
(TIF)

**S7 Fig. No induced correlations.**
(TIF)

**S8 Fig. Parameter recovery for Path Integration model.**
(TIF)

**S9 Fig. Parameter recovery for Kalman Filter model.**
(TIF)

**S10 Fig. Parameter recovery for Cue Combination model.**
(TIF)

**S11 Fig. Best fit parameters for the Hybrid model.**
(TIF)

**S12 Fig. Correlations between parameters in the Hybrid model.**
(TIF)

**S13 Fig. Comparison between data and model for all participants.**
(TIF)

**S14 Fig. Subject's confidence rating plotted against their angle error.**
(TIF)

**S15 Fig. Subject's confidence rating plotted against target location.**
(TIF)

**S16 Fig. Subject's confidence rating plotted against the posterior variance.**
(TIF)

**S1 File. Supplementary Material including mathematical derivation and fitting procedure.**
(PDF)

## Acknowledgments

The Authors are grateful to Joshua Dean Garren for helping during data collection and Tapas Jaywant Arakeri for helpful discussions.

## Author Contributions

**Conceptualization:** Sevan K. Harootonian, Robert C. Wilson.

**Data curation:** Sevan K. Harootonian.

**Formal analysis:** Sevan K. Harootonian, Robert C. Wilson.

**Funding acquisition:** Arne D. Ekstrom.

**Investigation:** Sevan K. Harootonian, Arne D. Ekstrom, Robert C. Wilson.

**Methodology:** Sevan K. Harootonian, Robert C. Wilson.

**Project administration:** Robert C. Wilson.

**Resources:** Arne D. Ekstrom.

**Software:** Sevan K. Harootonian.

**Supervision:** Arne D. Ekstrom, Robert C. Wilson.

**Validation:** Sevan K. Harootonian, Robert C. Wilson.

**Visualization:** Sevan K. Harootonian, Robert C. Wilson.

**Writing – original draft:** Sevan K. Harootonian, Arne D. Ekstrom, Robert C. Wilson.

**Writing – review & editing:** Sevan K. Harootonian, Arne D. Ekstrom, Robert C. Wilson.

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
