## [Decision Letter · Decision Letter 0]

2 Aug 2021

Dear Graduate Student Harootonian,

Thank you very much for submitting your manuscript "Combination and competition between path integration and landmark navigation in the estimation of heading direction" for consideration at PLOS Computational Biology.

As with all papers reviewed by the journal, your manuscript was reviewed by members of the editorial board and by several independent reviewers. In light of the reviews (below this email), we would like to invite the resubmission of a significantly-revised version that takes into account the reviewers' comments.

We cannot make any decision about publication until we have seen the revised manuscript and your response to the reviewers' comments. Your revised manuscript is also likely to be sent to reviewers for further evaluation.

Sincerely,

Joseph Ayers, PhD

Associate Editor

PLOS Computational Biology

Samuel Gershman

Deputy Editor

PLOS Computational Biology

Reviewer's Responses to Questions

**Comments to the Authors:**

Reviewer #1: Please see attached document.

Reviewer #2: This manuscript tested the contributions of body-based movement and visual landmarks during path integration. The authors set up an experiment where participants received visual feedback on an angular rotation, the feedback was either accurate (consistent with body information) or inconsistent, and the researchers compared how people used this feedback to integrate visual cues with body-based information. Comparing four potential models, they found that a Hybrid model – in which participants follow the combined visual/body information when the mismatch is small but follow body-based information when the mismatch is large – worked best.

Overall this is an impressive set of models and experimental work and it provides important insight into path integration and the combination of differing perceptual information during navigation. I had a few primary comments about the model comparison and some of the assumptions, and the remainder of the comments are about clarity.

1. For Figure 10 and the BIC model comparison, is there a cutoff number that indicates strongly in favor of the Hybrid model (akin to how the Bayes Factor scores are interpreted)? I would imagine that many of these are quite low and could be considered weak evidence. Currently the authors are taking anything that favors the Hybrid model as evidence, but that could be overinterpreting the findings.

2. Path integration model: This model assumes that everything that is happening is during the response phase. It sounds like it allows for error during encoding, which is the remembered angle. That’s fine if you are focusing just on the response portion, but you should probably make these kinds of assumptions explicit. For example, when reading the Appendix, it says “As they turn…” at first I thought this was for the encoding turn (or for both), but I think it’s just the response turn. I think the confusion arises because there are two processes: path integration and target comparison. Presumably you are integrating on the encoding turn as well, so that makes it a bit confusing here.

3. Intro: allocentric visual information is not just landmarks. Optic flow is a major allocentric visual cue to path integration, so that distinction should be made clear.

4. If the consistent feedback range is about 60 degrees, then for the inconsistent trials wouldn’t about 1/6 of the time in the random sampling the feedback was actually consistent? Does this affect the interpretation of the Hybrid model at all?

5. For models that incorporate path integration: How often is this path integration sampling occurring, or is it continuous? Mostly I’m looking at the schematic in Figure 2 (which is very helpful!), but wondering whether you are modeling this continuously even though the figure is discrete. Also, is the path integration always (leaky) and underestimating (as would be suggested by many models and experimental evidence), or is it random in its over and underestimates? The schematic suggests that it can be under and some points and over at others. Is this really what we see experimentally? This applies to the path integration sections of the other models.

6. The Discussion could talk a bit more about path integration models in general and how these results compare.

Minor concerns/clarification:

1. Figure 1: caption says 100 trials of FB, but the figure itself indicates 300 trials. Based on the text later on, I think the caption is reversed.

2. I don’t think it is specifically said, so the authors should make the prediction explicit in the path integration model that they expect no difference in response angle between feedback and no feedback conditions (per angle)

3. For some clarification, the Hybrid model says that it goes with either Path Integration or the Kalman Filter on a trial-by-trial basis? So any given trial is not diagnostic, but the collection of trials will tell you that it is the Hybrid?

4. For the combined estimate in Fig 4, where does this long tail come from? It doesn’t look Gaussian.

5. Figure 7, is the most negative slope person missing trials?

6. Figure 8 needs more caption. What do the counts mean?

7. Figure 11 needs more caption. What are the colors and filled and open dots?

8. Line 470: “future gold” should be “future goal”

Supplement:

9. For Table S1, I think the headings reflect older names, should have Hybrid and Kalman Filter listed. What does the check mark mean?

10. For the Bayesian decoding of target position, what is the basis for the assumption that people know their memory is imperfect and that they incorporate prior knowledge?

11. Would the possible alpha angles be related to what angles they had experienced in the experiment?

12. In between equation S40 and S41 (page S14) there seems to be a half sentence missing.

13. Page S17 the text says No Feedback Model but Figure S5 says Path Integration Model.

14. Page S18 the text says Cue Combination Model but Figure S6 says Hybrid Model.

15. Figure S13 caption – what is marked in yellow and what is green mean?

**Have the authors made all data and (if applicable) computational code underlying the findings in their manuscript fully available?**

Reviewer #1: Yes

Reviewer #2: Yes

PLOS authors have the option to publish the peer review history of their article (what does this mean?). If published, this will include your full peer review and any attached files.

Reviewer #1: No

Reviewer #2: No
---

## [Decision Letter · Decision Letter 1]

6 Jan 2022

Dear Graduate Student Harootonian,

We are pleased to inform you that your manuscript 'Combination and competition between path integration and landmark navigation in the estimation of heading direction' has been provisionally accepted for publication in PLOS Computational Biology. Please take a look at the few lingering comments from one reviewer and try your best to address them in the final version of the paper.

Best regards,

Joseph Ayers, PhD

Associate Editor

PLOS Computational Biology

Samuel Gershman

Deputy Editor

PLOS Computational Biology

Reviewer's Responses to Questions

**Comments to the Authors:**

Reviewer #1: Please see attached file.

Reviewer #2: It would be nice if some of the clarifications to the reviewer questions made it into the text itself, since likely others will have the same questions. Or just make the text clearer so that these confusions do not arise. But the authors have answered my questions satisfactorily and overall this is a very nice experiment and model.

**Have the authors made all data and (if applicable) computational code underlying the findings in their manuscript fully available?**

Reviewer #1: None

Reviewer #2: Yes

PLOS authors have the option to publish the peer review history of their article (what does this mean?). If published, this will include your full peer review and any attached files.

Reviewer #1: No

Reviewer #2: No

---

## [Editor Report · Acceptance letter]

7 Feb 2022

PCOMPBIOL-D-21-01111R1 

Combination and competition between path integration and landmark navigation in the estimation of heading direction

Dear Dr Harootonian,

I am pleased to inform you that your manuscript has been formally accepted for publication in PLOS Computational Biology. Your manuscript is now with our production department and you will be notified of the publication date in due course.

With kind regards,

Agnes Pap
